# Ability of a Set of Trunk Inertial Indexes of Gait to Identify Gait Instability and Recurrent Fallers in Parkinson’s Disease

**DOI:** 10.3390/s21103449

**Published:** 2021-05-15

**Authors:** Stefano Filippo Castiglia, Antonella Tatarelli, Dante Trabassi, Roberto De Icco, Valentina Grillo, Alberto Ranavolo, Tiwana Varrecchia, Fabrizio Magnifica, Davide Di Lenola, Gianluca Coppola, Donatella Ferrari, Alessandro Denaro, Cristina Tassorelli, Mariano Serrao

**Affiliations:** 1Department of Medico-Surgical Sciences and Biotechnologies, University of Rome Sapienza, 04100 Latina, Italy; stefanofilippo.castiglia@uniroma1.it (S.F.C.); dantetrabassi@uniroma1.it (D.T.); davide.dilenola@uniroma1.it (D.D.L.); gianluca.coppola@uniroma1.it (G.C.); 2Department of Human Neurosciences, University of Rome Sapienza, 00185 Rome, Italy; antonella.tatarelli@uniroma1.it (A.T.); fabrizio.magnifica@uniroma1.it (F.M.); 3Department of Occupational and Environmental Medicine, Epidemiology and Hygiene, INAIL, Monte Porzio Catone, 00078 Rome, Italy; a.ranavolo@inail.it (A.R.); t.varrecchia@inail.it (T.V.); 4Movement Analysis Research Unit, IRCCS Mondino Foundation, 27100 Pavia, Italy; rob.deicco@gmail.com (R.D.I.); valentina.grillo@mondino.it (V.G.); cristina.tassorelli@unipv.it (C.T.); 5Department of Brain and Behavioral Sciences, University of Pavia, 27100 Pavia, Italy; 6U.O. Riabilitazione Neurologica, Istituto Chirurgico Ortopedico Traumatologico, Latina 04100, Italy; donat.ferrari@tiscali.it (D.F.); alessandrodenaro@hotmail.com (A.D.); 7Movement Analysis Laboratory, Policlinico Italia, 00162 Rome, Italy

**Keywords:** gait disorders, neurologic, accelerometry, Parkinson’s disease, harmonic ratio, recurrence quantification analysis, falls, postural balance

## Abstract

The aims of this study were to assess the ability of 16 gait indices to identify gait instability and recurrent fallers in persons with Parkinson’s disease (pwPD), regardless of age and gait speed, and to investigate their correlation with clinical and kinematic variables. The trunk acceleration patterns were acquired during the gait of 55 pwPD and 55 age-and-speed matched healthy subjects using an inertial measurement unit. We calculated the harmonic ratios (HR), percent recurrence, and percent determinism (RQAdet), coefficient of variation, normalized jerk score, and the largest Lyapunov exponent for each participant. A value of ≤1.50 for the HR in the antero-posterior direction discriminated between pwPD at Hoehn and Yahr (HY) stage 3 and healthy subjects with a 67% probability, between pwPD at HY 3 and pwPD at lower HY stages with a 73% probability, and it characterized recurrent fallers with a 77% probability. Additionally, HR in the antero-posterior direction was correlated with pelvic obliquity and rotation. RQAdet in the antero-posterior direction discriminated between pwPD and healthy subjects with 67% probability, regardless of the HY stage, and was correlated with stride duration and cadence. Therefore, HR and RQA_det_ in the antero-posterior direction can both be used as age- and-speed-independent markers of gait instability.

## 1. Introduction

Gait abnormality is one of the most invalidating features of persons with Parkinson’s disease (pwPD) [1] and closely parallels disease progression [2], leading to a high risk of falls [2], impaired patient autonomy, and reduced quality of life [3,4]. Based on the significant physical and economic burdens associated with falls [5,6], it is critical to identify sensitive and specific indexes that can represent various aspects of gait imbalance [7]. This would allow us to improve the ability to assess gait stability in pwPD, providing a multifold positive impact in terms of patient care ranging from the prevention of falls to the possibility of monitoring and optimizing both pharmacological and rehabilitation treatments [8,9]. Although many clinical tools have been developed to identify unbalance in pwPD [10,11,12], technology-based objective measures may improve the ability to capture motor behaviors to optimize treatment strategies [8,13].

From a biomechanical perspective, gait stability refers to the ability to maintain functional locomotion in the presence of small kinematic disturbances or control errors [14,15,16,17]. A series of quantitative biomechanical measures have been proposed to assess dynamic gait imbalance in pwPD [18]. One of the most widely explored measures is the coefficient of variation (CV) of spatio-temporal parameters (i.e., stride time and step length) [19,20,21,22,23,24]. Other reported measures have been derived from the accelerative patterns of the trunk and include the harmonic ratio (HR) [25,26], largest Lyapunov exponent (LLE) [27] and normalized jerk score (NJS) [28,29]. Another trunk-acceleration-derived measure, the recurrence quantification analysis (RQA), has been reported to be impaired in patients with vestibulopathy [30] and provides a promising approach to the investigation of center of pressure fluctuations in older fallers [31].

In almost all previous studies, gait stability indexes have been investigated individually to identify preventive measures for falls. However, in general terms, it would be ideal to obtain a set of gait stability indexes to understand which are the most sensitive in terms of distinguishing between pathological and neurotypical gaits, which are the most closely related to the severity of the disease, which aspects of stability are expressed by various indexes, which are the most useful for predicting falls, and which are the most responsive to rehabilitation treatments [7].

Additionally, because the correlations between gait speed, age, and trunk accelerations have been well described and because walking stability is strongly influenced by gait speed in pwPD [32], which is a marker of gait alteration common to most neurological diseases [33], the value of the gait stability indexes as independent markers of gait alteration should be investigated, regardless of these confounding factors [33].

Based on their ease of use and cost effectiveness, wearable sensors are widely used for clinical assessment to obtain more objective measures of walking performance [34,35,36,37,38,39]. Based on the inverted pendulum model of gait [40], mobile inertial measurement units (IMU), which embed accelerometers and gyroscopes, are able to objectively capture the ability to control the body center of mass while moving the base of support, resulting in an effective tool to monitor dynamic balance during gait [36,41]. Ideally, by using several combined IMUs to analyze a subject’s gait, the overall accuracy of the analysis would improve, but this gain would be offset by the wearability burden. Conversely, a single lumbar-mounted IMU provides sensitive information on the gait of pwPD and allows clinicians to monitor the gait of pwPD also in free-living conditions [42]. IMUs directly provide trunk acceleration measures and facilitate the recording of patient gaits for many steps during follow-up clinical assessments in outpatient facilities, making them ideal tools for investigating gait stability in pwPD.

In this study, we measured a set of gait stability indexes in data samples from pwPD that were collected using a wearable device with the goals of (i) determining the accuracy of each index in terms of detecting gait instability in pwPD compared to healthy subjects (HS) by excluding the effects of age and gait speed between groups, (ii) assessing the ability of each identified index to characterize the gait of pwPD who fall recurrently, compared with pwPD who do not fall recurrently, and (iii) exploring the correlations of each index with clinical and biomechanical variables. Our main hypotheses can be summarized as follows: (i) Some gait stability indexes may have sufficient ability to discriminate between pwPD and HS and may be correlated with the spatio-temporal and kinematic gait features of pwPD, regardless of gait speed; (ii) the identified discriminant indexes may identify frequent fallers among pwPD as an expression of walking instability. 

## 2. Materials and Methods

### 2.1. Subjects

We collected data samples from 62 subjects diagnosed with idiopathic PD (20 females and 42 males, aged 73 ± 6.60 years) who enrolled for our study at “ICOT” in Latina, Italy, and the “IRCCS Mondino Foundation” in Pavia, Italy. The inclusion criteria were defined as follows: (i) diagnosis of idiopathic PD according to the UK bank criteria [43], (ii) Hoehn and Yahr (HY) stages 1 to 3 [44], (iii) ability to walk independently for at least 8 m along a laboratory pathway without exhibiting gait freezing and the ability to perform repeated walking trials with at least 10 consecutive strides [45,46,47], and (iv) on a stable drug program and acclimated to their current medication use for at least two weeks. The exclusion criteria were defined as follows: (i) cognitive deficits (defined as scores less than 26 on the Mini-Mental State Examination) [48,49], (ii) moderate or severe depression (defined as scores greater than 17 on the Beck Depression Inventory) [50,51], and (iii) orthopedic and/or other gait-influencing diseases, such as other neurological diseases, clinically defined osteoarthritis [52,53,54], or total hip joint replacement. Particularly, subjects referring pain in hip or knee joints and reduced range of motion on internal hip rotation, hip flexion and visible anatomic alterations of the joints were excluded [54].

For group comparisons, a group of 55 HS (aged 61.40 ± 11.98 years) were recruited. Each HS repeated the tests twice: once walking at a self-selected speed and once walking at a slower directed speed.

Informed consent was obtained from all participants in compliance with the Helsinki Declaration and local ethics committee approval was obtained (CE Lazio2, protocol number: 0053667/2021).

### 2.2. Instrumentation

An inertial sensor (BTS GWALK, BTS, Milan, Italy) attached to an ergonomic belt placed around the pelvis at the level of the L5 vertebra and connected to a laptop via Bluetooth was used to acquire data. This sensor is equipped with a tri-axial accelerometer (16 bit/axes), tri-axial magnetometer (13 bits), and tri-axial gyroscope (16 bit/axes). Linear trunk accelerations and trunk angular velocities in the anterior-posterior (AP), mediolateral (ML), and vertical (V) directions were measured at a sampling rate of 100 Hz (Figure 1).

### 2.3. Task Description

To familiarize themselves with the experimental procedure, the participants were asked to walk on the ground along a predetermined pathway before the experimental session. Both the pwPD and HS were asked to walk barefoot at a self-selected speed along a corridor (approximately 3 m wide and 10 m long). To avoid the influence of gait speed on other gait parameters and collect the largest possible sample size for speed-matched comparisons, the HS were also asked to walk at a slower speed [1,55,56,57,58,59]. Because natural locomotion was the main focus of this study, only general and qualitative instructions were provided, leaving the subjects free to choose their speed level without interfering with their pacing and rhythm based on external sensory cues.

### 2.4. Inertial Sensor Data Processing

The “Walk+” protocol of the G-STUDIO software (G-STUDIO, BTS, Milan, Italy) was used to detect trunk acceleration patterns, right and left heel strikes, toe-off, spatiotemporal parameters, and pelvis kinematics. The HR, RQA, CV, NJS, and largest Lyapunov exponent (LLE) were calculated (Figure 1) using MATLAB (MATLAB 7.4.0, MAthWorks, Natick, MA, USA).

#### 2.4.1. HR Calculation

The HR [60] provides an indication of the acceleration patterns, smoothness, and rhythm of the gait. The greater the HR value, the smoother the walking pattern.

A stable and rhythmic gait pattern should be characterized by multiples of two acceleration patterns within any given stride. Irregular acceleration signals during a walking trial reflect unstable gait patterns and should be characterized by acceleration patterns that do not repeat in multiples of two, resulting in out-of-phase accelerations. The harmonic content of the trunk acceleration signals was analyzed in the AP (HR_AP_), ML (HR_ML_), and V (HR_V_) directions by using stride frequency as the principal frequency component. Because most of the power in a normal cadence occurs below 10 Hz, 20 harmonics were calculated for each subject according to their stride times. A discrete Fourier transform was used to break down the trunk accelerations of each stride into individual sinusoidal waveforms. In a stable and smooth gait pattern, the acceleration signals in the AP and V directions repeat in multiples of two during a single stride. Therefore, HR_AP_ and HR_V_ were calculated as the ratio of the sum of the amplitudes of the first 10 even harmonics divided by the sum of the amplitudes of the first 10 odd harmonics. HR_ML_ was calculated as the sum of the amplitudes of the odd harmonics divided by the sum of the amplitudes of the even harmonics. A high-pass filter with a 20 Hz cutoff was used to eliminate noise signals and the HR values were calculated as follows [61]:$$HR_{AP,\;V\;}=\frac{\sum_{i}A_{i*2}}{\sum_{i}A_{i*2-1}},$$
$$HR_{ML}{}_{\;}=\frac{\sum_{i}A_{i*2-1}}{\sum_{i}A_{i*2}},$$
where $A_{i}$ represents the amplitudes of the first 20 even harmonics and *A*_2*i*−1_ represents the amplitudes of the first 20 odd harmonics.

#### 2.4.2. RQA Calculation

RQA is a nonlinear technique that can provide useful information regarding the patterns and structures of system dynamics. It provides a characterization of a variety of features of a given time series, including the quantification of deterministic structures and non-stationarity, based on the construction of recurrence plots. A detailed description of RQA calculation was provided by Webber et al. (1994) [62]. Acceleration and angular velocity data are embedded in “m” dimensions using “m” copies of the original time series, where each copy is shifted in time by integer multiples of “τ” samples. The embedding dimension “m” is the first recurrence parameter and is estimated using the nearest-neighbor method [63], which compares the distances between neighboring trajectories at successively higher dimensions. ‘‘False neighbors’’ occur when trajectories that overlap in dimension $m_{i}$ are distinguished in dimension $m_{i+1}$. As *i* increases, the total percentage of false neighbors decrease, and *m* is chosen where this percentage approaches 0. False neighbors analysis was performed using values of Rtol = 17 and Atol = 2 [14,30,46,64,65]. We considered a maximum embedding dimension of 10 (range: 2–10), and m = 5 was considered as the optimal embedding dimension. Here, “τ” represents the second recurrence parameter and is selected to minimize the interaction between the points in the measured time series. Two common methods for selecting a proper delay are finding the first minimum in either the (linear) autocorrelation function or (nonlinear) mutual information function of the continuous time series. τ was calculated from the first minimum of the average mutual information (AMI) function [66], which evaluates the shared amount of information in bits between two data sets over a range of time delays. By choosing the first minimum of the AMI function, adjacent delay coordinates with a minimum of redundancy are provided. A range of τ between 7 and 18 was evaluated in this study. The time delay computed by the first minimum of the AMI considered as optimal was 10 samples [46,62,64,65].

A distance matrix is then computed by calculating the Euclidean distances between all embedded vectors. A recurrence matrix is computed by selecting a threshold (radius) of 10% of the maximum distance, where all cells with values below this threshold are identified as recurrent points. RQA variables are used to quantify the structure of the recurrence matrix. Percent recurrence (RQA_rec_) can be calculated to understand how often a trajectory visits similar locations in the state space (time independent) and is computed as the percentage of recurrent points in the recurrence matrix. Percent determinism (RQA_det_) can be calculated to understand how often a trajectory repeatedly revisits similar state space locations (time dependent) and is quantified as the percentage of recurrent points in the diagonal line structures (at least four consecutive points in length) parallel to the main diagonal. Therefore, RQA_rec_ quantifies the number of potentially recurrent points, where only a portion of these points recur periodically and are related to the predictability of the target dynamical system. The higher the RQA_rec_ and RQA_det_ values, the higher the predictability of the system [30].

#### 2.4.3. CV

To compute the step length CV, the step length was estimated from the upward and downward movements of the trunk, as proposed by Zijlstra and Hof [67]. Assuming a compass gait type, the body’s center of mass (CoM) movements in the sagittal plane follow a circular trajectory during each single-support phase. In this inverted pendulum model, changes in the height of the CoM depend on the step length [67]. Therefore, the step length is calculated as follows:$$step\;length=2\sqrt{2lh-h^{2}},$$
where *h* denotes the change in the height of the CoM and *l* represents the pendulum length. A double integration of the vertical acceleration was implemented to calculate changes in the vertical position. A fourth-order zero-lag Butterworth high-pass filter (0.1 Hz) was used to avoid integration drift. The amplitude of the changes in the vertical position (h) was calculated as the difference between the highest and lowest positions during a step cycle. The leg length was considered to be the pendulum length (l). The step length was calculated as the mean of the observed step lengths during seven consecutive steps for each subject. The stride length CV was computed as follows:
*CV* = 100*SD*/*mean*,
where *mean* is the mean step length and *SD* is the standard deviation over all step lengths for each subject [68]. The higher the *CV*, the higher the variability in step length.

#### 2.4.4. NJS

The NJS measures the time-normalized rate of change in the acceleration signal during stepping [28,29]. The acceleration data were first bandpass filtered using a fourth-order zero-lag Butterworth filter with a cutoff frequency of 20 Hz. We then calculated the NJS from the time duration between each foot contact by applying the following formula [29]:$$Gait\;NJS=\frac{1}{N}\sum_{i=1}^{N}\sqrt{\frac{{\left(hs_{i+1}-hs_{i}\right)}^{5}}{2}}\int_{hs_{i}}^{hs_{i+1}}{\left(a\right)}^{2}dt,$$
where *hs_i_* is the time of the *i*th heel strike and *a* is the acceleration. The next step consists of bandpass filtering of the NJS using a fourth-order zero-lag Butterworth filter with a cutoff frequency of 5 Hz [28]. The higher the NJS, the greater the gait smoothness.

#### 2.4.5. LLE

The LLE characterizes the behavior of a dynamical system. It quantifies gait stability as the average logarithmic rate of divergence following infinitesimal perturbations. An LLE less than zero represents the rate of convergence of the system’s trajectory to its nearest neighboring trajectory, whereas an LLE greater than zero represents the rate of divergence. When trajectories converge, the observed system is considered to have local dynamic stability, whereas divergence indicates local dynamic instability. The procedure described by Van Schooten et al. (2014) [69] was used to estimate the LLE in our study. To avoid the loss of spatiotemporal fluctuations and nonlinearities, no filtering was applied to the triaxial accelerations and the accelerations were time-normalized to obtain 100 data points per stride, thereby excluding the effects of the time duration of the data series on dynamic stability measures. Local dynamic stability was computed for each triaxial trunk acceleration over the strides considered in each trial. We assessed the short-term maximum finite-time LE $(\lambda_{max}$) for each stride based on the AP, ML, and V trunk accelerations. The value of $\lambda_{max}$ was determined according to Rosenstein’s algorithm for short time series by using the Lyaprosen MATLAB toolbox for nonlinear time series analysis. A multidimensional state space, whose dimensions were determined using the classical global false-nearest-neighbor method, was reconstructed from the recorded one-dimensional time series data by juxtaposing the original data and delayed copies (the time delay was determined according to the first minimum of the average mutual information function) to evaluate dynamic perturbations [70]. Low values of this index indicate more stable trunk dynamics and high values indicate less stable trunk dynamics.

### 2.5. Clinical Assessment

The severity of PD was assessed using the HY disease staging system and the motor examination section of the Unified Parkinson’s Disease Rating Scale (UPDRS-III) [71]. Clinical scales were administered by an assessor unaware of the stability index reports. To identify fallers among the pwPD, participants were also asked to report the number of falls they experience over the past year.

### 2.6. Statistical Analysis

A needed sample of 62 subjects (31 pwPD and 31 HS) was calculated to identify gait stability indexes with good ability (area under the receiver operating characteristic (ROC) curve (AUC ≥ 0.70) for discriminating between pwPD and HS at a 95% significance level with 80% power under the null hypothesis of AUC ≤ 0.50. Based on the assumption of a high dropout rate as a result of the number of recorded strides required to be included in the analysis and matching procedures, we initially recruited 117 participants.

A one-to-one optimal data matching procedure using the propensity score difference method was implemented to match pwPD with HS [72]. We excluded 21 out of the 110 recorded HS trials based on an insufficient number of recorded strides. Therefore, 89 HS trials were used for the matching procedure. Propensity scores were calculated for each HY subgroup through logistic regression analysis using age [73,74] and speed [32,75,76,77] as covariates. Univariate analysis of variance (ANOVA) with Dunnett’s post-hoc analysis [78] was performed to verify the effectiveness of the age and speed matching procedures between pwPD at each HY stage and matched HS (HS_matched_).

After checking the normality of the distributions and homoscedasticity of the variances using the Shapiro-Wilk test for all variables and Levene’s test, respectively, univariate ANOVA with Bonferroni’s post-hoc analysis and the Kruskal-Wallis H test with Dunn’s post-hoc analysis was performed to identify differences in the gait stability indexes between pwPD and HS_matched_, as well as across the HY stages within the pwPD.

To assess the ability of the identified indexes to discriminate between pwPD and HS_matched_, characterize the gait of pwPD according to the HY stages, and characterize the gait of recurrent fallers, ROC curves were plotted and AUCs were calculated with the presence of PD, HY stages, and a reported number of falls ≥5 as anchors [79,80,81], respectively. AUC values greater than 0.60 with a confidence interval lower bound greater than 0.50 were considered for sufficient overall discriminative ability [82]. The optimal cutoff points (OCPs) for each discriminative index were calculated as the points on the ROC curves that maximized the sum of sensitivity (Se) and specificity (Sp) values. Positive and negative likelihood ratios (LR+ and LR−) were also computed and transformed into post-test probabilities by using Fagan’s nomogram to analyze the probability of being correctly classified by a given index at the OCP [83]. Because the prevalence of recurrent fallers in our sample was low, to improve the generalizability of our results, the previously reported 39% prevalence of recurrent fallers in the general PD population [80,84] was used to calculate post-test probabilities for the identification of recurrent fallers [85].

Partial correlation coefficients (r) after removing the effects of age and gait speed were calculated to identify correlations between the identified indexes and UPDRS-III scores, history of falls, and spatiotemporal and kinematic parameters. Statistical analyses were performed using the IBM SPSS ver. 27 and NCSS 2019 software.

## 3. Results

### 3.1. Subjects and Matching Procedure Results

After the matching procedure, 55 walking trials from pwPD and 55 age-and-speed-matched walking trials from 30 HS (HS_matched_) were included in our analysis. The final pwPD group consisted of 16 females and 39 males diagnosed with PD 8.20 ± 5.37 years ago and aged 71.17 ± 4.74 years. Twelve subjects were assessed at the HY = 1 disability stage, 19 subjects at the HY = 2 stage, and 24 at the HY = 3 stage. The mean UPDRS-III score was 39.48 ± 16.91. Nineteen subjects (34.4%) had experienced at least one fall in the previous year with 1.63 ± 4.54 falls an average. Seven subjects (12.7%) reported more than five falls during the past six months and were classified as recurrent fallers [79]. The mean gait speed of the included pwPD was 0.76 ± 0.23 m/s. The final HS_matched_ group consisted of 55 walking trials from a sample of 30 subjects (15 females and 15 males aged 70.11 ± 5.62 years). Eighteen of the 55 HS_matched_ trials were recorded at self-selected speeds (0.93 ± 0.38 m/s) and 37 were recorded at a reduced speed (0.65 ± 0.17 m/s). The mean speed of the final HS_matched_ group was 0.77 ± 0.25 m/s.

No significant overall differences between group means were identified for age (F*_(3, 105)_* = 2.81; *p* = 0.29) or gait speed (F*_(3, 106)_* = 2.49; *p* = 0.06) after the matching procedure. Post-hoc analysis revealed no significant differences for age (HY=1 vs. HS: *p* = 1.00; HY = 2 vs. HS: *p* = 1.00; HY = 3 vs. HS: *p* = 0.09; HY = 1 vs. HY = 2: *p* = 1.00; HY = 1 vs. HY = 3: *p* = 0.11; HY = 2 vs. HY = 3: *p* = 0.25) and gait speed (HY=1 vs. HS: *p* = 1.00; HY = 2 vs. HS: *p* = 1.00; HY = 3 vs. HS: *p* = 0.29; HY = 1 vs. HY = 2: *p* = 1.00; HY = 1 vs. HY = 3: *p* = 0.12; HY = 2 vs. HY = 3: *p* = 0.21) between the HY stage subgroups and HS, neither between the HY subgroups.

### 3.2. Discriminative Ability Results

Significant main effects of the group on HR_AP_ (H = 9.48, *p* = 0.024), HR_ML_ (H = 10.24, *p* = 0.017), HR_V_ (H = 14.30, *p* = 0.003), RQA_det_ in the AP direction (RQA_detAP_) (H = 8.27, *p* = 0.041), and CV (H = 10.41, *p* = 0.015) were identified (Figure 2, Table 1).

Post-hoc analysis revealed significant differences in HR_AP_ between pwPD at HY stage = 3 and HS_matched_ (*p* = 0.003), between pwPD at HY = 3 and HY = 1 (*p* = 0.043), and between pwPD at HY = 3 and HY = 2 (*p* = 0.032). A significant difference was identified in HR_ML_ between pwPD at HY = 3 and HS_matched_ (*p* = 0.001). Significant differences were identified in HR_V_ between pwPD at HY = 3 and HS_matched,_ (*p* = 0.000)_,_ and between pwPD at HY = 3 and HY = 2 (*p* = 0.037). Significant differences were identified in RQA_detAP_ between pwPD at all HY stages and HS_matched_ (HY = 1, *p* = 0.015; HY = 2, *p* = 0.041, and HY = 3, *p* = 0.039). A significant difference in CV was identified between pwPD at HY = 3 and HS_matched_ (*p* = 0.004) (Table 1).

A good ability (AUC > 0.70) to discriminate between pwPD at HY = 3 and HS_matched_ was identified for HR_AP_, HR_ML_, HR_V_, CV (Table 2, Figure 3 and Figure 4). A moderate ability (AUC = 0.65) to discriminate between pwPD at HY = (1, 2, 3) and HS_matched_ was identified for RQA_detAP_ (Table 2, Figure 4). HR_AP_ values ≤ 1.50 identified pwPD at HY = 3 with 67% probability, HR_ML_ values ≤ 1.58 identified pwPD at HY = 3 with 54% probability, HR_V_ values ≤ 1.74 identified pwPD at HY = 3 with 57% probability, and CV values ≥ 38.06 identified pwPD at HY = 3 with 58% probability. RQA_detAP_ values ≤ 38.85 identified pwPD with 67% probability, regardless of the HY stage.

HR_AP_ exhibited a good ability to discriminate between pwPD at HY = 3 and HY = (1, 2) (AUC = 0.70) (Table 2, Figure 3). HR_AP_ values ≤ 1.50 discriminated pwPD at HY = 3 from subjects at lower HY stages with 73% probability. Furthermore, HR_AP_ exhibited a strong ability to identify recurrent fallers (AUC = 0.80) (Table 2, Figure 3). HR_AP_ values ≤ 1.50 identified frequent fallers with 77% probability.

### 3.3. Partial Correlation Analysis Results

After removing the effects of gait speed and age, HR_AP_ was negatively correlated with the history of falls (r = −0.45, *p* = 0.004) and positively correlated with pelvic obliquity (r = 0.37, *p* = 0.024) and pelvic rotation (r = 0.31, *p* = 0.040). RQA_detAP_ was negatively correlated with gait cadence (r = −0.35, *p* = 0.031) and positively correlated with stride time (r = 0.36, *p* = 0.030) and UPDRS III score (r = 0.385, *p* = 0.004). HR_ML_, HR_V_, and CV exhibited no correlations with clinical features or gait kinematics.

## 4. Discussion

The main goal of this study was to determine the ability of 16 gait stability indexes derived from trunk acceleration signals to differentiate between pwPD and HS. By excluding the effects of age and gait speed, we found that the HRs calculated for the three spatial planes (AP, ML, and V) and the step length CV were able to discriminate between pwPD at moderate stages of disease progression (HY = 3) and HS with 67%, 54%, 57%, and 58% probabilities, respectively. HR_AP_ was also able to discriminate between pwPD at the moderate disability stage (HY = 3) and pwPD at lower disability stages with 73% probability. Furthermore, HR_AP_ exhibited good ability to characterize the gait of recurrent fallers with 77% probability. RQA_detAP_ was able to discriminate between pwPD and HS_matched_, regardless of the HY score, with 67% probability. Overall, these results are in line with those from previous studies reporting that pwPD exhibit disruptions in the rhythmicity of pelvic acceleration [25,26] and a divergence from the typical quasi-periodic time-dependent recurrence of gait [86,87], particularly in the AP direction.

HR is a gait index that quantifies the smoothness of the trunk acceleration patterns in the three spatial directions during the gait and is defined as the ability to synchronize changes in trunk mechanics with those in the lower extremities [18]. In this study, the ability to control the trunk smoothly during the gait was found to be altered in pwPD at HY = 3. In particular, the HR in the AP direction was found to be correlated with a decrease in pelvic mobility. Additionally, we found that HR_AP_ differentiated pwPD according to disease stages and was correlated with the history of falls, exhibiting a strong ability to identify recurrent fallers. Considering the progression of PD, HY stage 3 classifies subjects with PD with mild bilateral and axial involvement and initial postural instability. Therefore, HR_AP_ may reflect the appearance of postural instability leading to falls and is correlated with axial rigidity. In this study, we could not directly measure trunk kinematics based on the localization of the inertial measurement unit. Therefore, information regarding lower trunk kinematics was inferred from pelvic kinematics [35]. In particular, the correlation coefficients revealed that HR_AP_ was lower in subjects with lower pelvic obliquity and pelvic rotation. Alterations in trunk rotation and pelvic kinematics have been consistently described as characteristics of pwPD [55,88,89]. Based on our results, we can argue that the trunk rotation rigidity reflected by pelvic rotation and pelvic obliquity leads to alterations in AP trunk smoothness during the gait. This deficit in AP trunk control may be considered as a marker of postural instability that is independent of age and gait speed, as confirmed by our partial correlation analysis with a history of falls.

Although we determined that the HR in the ML and V directions can differentiate between pwPD at HY = 3 and HS_matched_, the low discriminative power (<60%) of these variables for correctly identifying pwPD and their lack of correlations with clinical features disqualify them as accurate markers of gait instability in PD. These results are in line with previous research [25,26] evaluating the ability of HR_ML_ [26] and HR_V_ [25] to differentiate between pwPD and HS, yet revealing a loss in their informative value when adjusting for gait speed [25,26]. Additionally, we did not find any correlations between HR_ML_ or HR_V_ and a history of falls. In contrast, Cole et al. (2017) [32] identified significant differences in HR values in all planes between pwPD and age-matched healthy subjects, as well as a strong association between the HR values of recurrent fallers and gait speed. However, in their study, the experimental groups were not matched by gait speed. Therefore, direct comparisons to our results are difficult. The objective of this study was to determine the discriminative ability of HR values irrespective of gait speed. Therefore, our results indicate that HR_ML_ and HR_V_ cannot be considered as independent markers of falling risk.

RQA_det_ is an index that expresses the predictability of acceleration trajectories during the gait. It represents how often a subject’s trunk accelerations revisit similar locations in the three spatial planes during their gait. Previous studies [86,87,90] have shown that RQA_det_ is the most convenient RQA index for objectively separating pwPD from HS. Accordingly, we identified lower RQA_det_ values in the AP direction in pwPD compared to HS_matched_, regardless of the disability stage. Therefore, we can argue that pwPD exhibit disruptions in the quasi-periodic recurrence of their gaits in the early stages of PD, which reflects the early temporal gait alterations experienced by pwPD [91]. This consideration is further reinforced by the correlation we identified between RQA_detAP_ and temporal gait parameters, such as cadence and stride duration. Since the earliest stages of PD, pwPD exhibit shorter step lengths and stride times with an increased cadence [55,92] as a part of the abnormal gait pattern that progressively deteriorates into festination [7]. The reduced predictability of AP trunk accelerations may reflect systemic instability caused by altered temporal gait patterns. Therefore, RQA_detAP_ can be considered as a temporal marker of gait stability that can identify pwPD independently of gait speed. Furthermore, RQA_detAP_ was the only index correlated with the motor aspects of PD measured using UPDRS-III scores. Considering the variety of motor symptoms other than gait impairment assessed by UPDRS-III scores, a lack of correlation with gait-oriented measures is reasonable [93]. Therefore, the positive correlation between RQA_detAP_ and UPDRS-III may reflect its ability to capture gait impairment besides other symptoms such as reduced hand dexterity, facial mimicry, or altered posture.

The step length CV is a measure of gait spatial variability. Significant differences in the step length CV between pwPD and HS have been observed by other researchers [18,74,94]. These alterations have been attributed to compensatory behaviors based on the reduction of arm swing or to expressions of problems in terms of regulating the amplitude of step length during walking [95]. Regarding HRs, we found specific alterations in the step length CV only in pwPD with HY = 3. This result is in line with the results of previous studies that did not observe step length CV alterations in subjects at milder disease stages [94]. We speculate that in terms of HRs, CV alterations occur concomitantly with the progression of bilateral and axial involvement. However, the low probability of correctly discriminating pwPD (58%) at the cutoff value and a lack of correlations with clinical variables and spatiotemporal gait parameters do not allow us to consider this index as a marker of gait stability that is independent of speed.

Previous studies have reported [29,96] significantly lower NJS values in the ML and V directions in pwPD compared to age-matched HS. These differences have been interpreted as a reflection of PD-related bradykinesia, which parallels lower gait speed [96] and reduced arm swing movements [97]. Furthermore, the NJS in the ML and V directions has been reported to be responsive to dopaminergic medication in subjects with improved gait speed. In this study, we identified no differences in NJS between pwPD and age-and speed-matched HS, indicating that NJS is a marker of a general loss in complexity of the motor control system [29] that depends on gait speed.

We also identified no differences between groups in the LLE. In contrast, other studies have reported significant group differences between pwPD and HS [59] and between walking at self-selected speeds and dual-task walking [98] in terms of LLE. These contrasting results can be attributed to varying matching procedures and testing conditions. In one of these studies [98], no speed matching was performed between pwPD and HS because they were focused on observing changes in local dynamic stability induced by dual-task conditions, which does not correspond to our aim. Another study [59] observed differences when walking at comfortable speeds between pwPD and non-matched HS, as well as an increase in local dynamic instability when speeding up the gait of pwPD to match the HS gait speed. Given that the goals of the aforementioned studies were different, combining their interpretations with our results may suggest that the gait of pwPD is characterized by dynamic instability, but that LLE is speed dependent.

One of the main strengths of this study is that we investigated the discriminatory ability of the target indexes as independent diagnostic tools by excluding the influence of one of the gait features that differentiates pwPD from HS at the same age, namely gait speed [7]. Therefore, combined with the post-test probability results, the relatively low AUC values we observed may suggest that the identified indexes cannot be used as diagnostic markers for PD. However, these indexes may represent useful tools for assessing subtle gait alterations that clinical observations cannot detect. Furthermore, our results demonstrated that HR_AP_ is a marker of gait alteration that can potentially identify recurrent fallers. Falls at a rate of more than one per year are frequently reported by elderly people [80]. The ability to detect recurrent fallers represents a key feature that a gait stability index should address because it can identify subtle alterations leading to falls in pwPD when compared to HS [80]. However, these results should be interpreted with caution. In this study, we only investigated gait patterns during the steady state. In pwPD, most falls occur during transitions such as turning or during the execution of concomitant cognitive tasks [84]. We attempted to overcome this limitation by using a high number of reported falls (greater than five) [80] as an anchor to discriminate recurrent fallers. Therefore, we can argue that HR_AP_ is able to characterize the gait of recurrent fallers among the most disabled subjects, even during the analysis of linear walking without additional tasks. Regardless, the prevalence of fallers in our sample was lower than that in the general PD population, which can be attributed to our inclusion criteria. Because many predictors of falls, such as cognitive deficits, freezing of gait, depression, and orthopedic and peripheral conditions, were considered as exclusion criteria, this may have led to a lower prevalence of potential fallers, which could influence the calculation of post-test probabilities [85]. We compensated for this limitation by using the reported 39% prevalence of recurrent fallers in the general PD population in our calculations of post-test probabilities, thereby increasing the generalizability of our results to larger samples. Another limitation of this study is the retrospective self-reported history of falls, which could have led to recall bias. Therefore, additional longitudinal studies investigating the ability of HR_AP_ to predict falls during the gait and more complex tasks should be conducted. Noteworthy, other acceleration-derived gait stability indexes, such as entropy measures or the maximal longest diagonal in recurrence quantification analysis, which we did not consider in this study, can be calculated to assess human gait instability. However, as these indexes are highly dependent on the number of recorded consecutive strides, other studies involving long-time gait series recordings are needed to analyze their ability to characterize the gait instability of pwPD.

## 5. Conclusions

In conclusion, the results of this study demonstrated that HR_AP_ and RQA_detAP_ are age- and speed-independent markers of gait instability in subjects with PD. Both HR_AP_ and RQA_detAP_ are able to discriminate pwPD from HS matched for age and gait speed. HR_AP_ is altered in subjects at HY = 3, correlated with alterations in pelvic movements, and are able to characterize the gait instability of recurrent fallers. RQA_detAP_ captures early temporal gait alterations in pwPD but is not able to characterize the gait of fallers. Given the ease of use and relatively low cost of inertial motion sensors, these indices could provide useful clinical information regarding gait instability or alterations in the pace of subjects with PD, which are not directly observable through routine clinical assessments. Additional studies should be conducted to assess their ability to predict future falls, identify alterations during more demanding gait tasks, and their responsiveness to pharmacological and rehabilitative treatments.

## Figures and Tables

**Figure 1 sensors-21-03449-f001:**
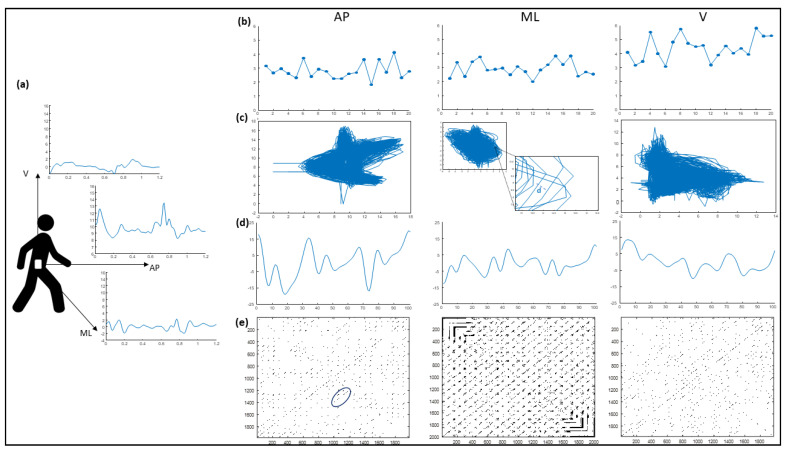
Graphical representation of the accelerations-derived gait indexes of a representative healthy subject: (**a**) amplitudes of the filtered acceleration signals in the antero-posterior (AP), medio-lateral (ML), and vertical (V) direction as a function of the time range data; (**b**) Harmonic Ratio values for each of the 20 considered strides; (**c**) 2D-reconstructed state space of the acceleration and its time-delayed copies (time delay of 12 data samples). The distance (***d***) of two neighboring trajectories at a one-time sample, which is needed to calculate the Lyapunov exponent, is highlighted; (**d**) representation of the jerks during the whole gait cycle; (**e**) recurrence matrix. Based on the percent of the recurrent points in the diagonal line structure parallel to the main diagonal (i.e., the blue circled points), the RQAdet was calculated.

**Figure 2 sensors-21-03449-f002:**
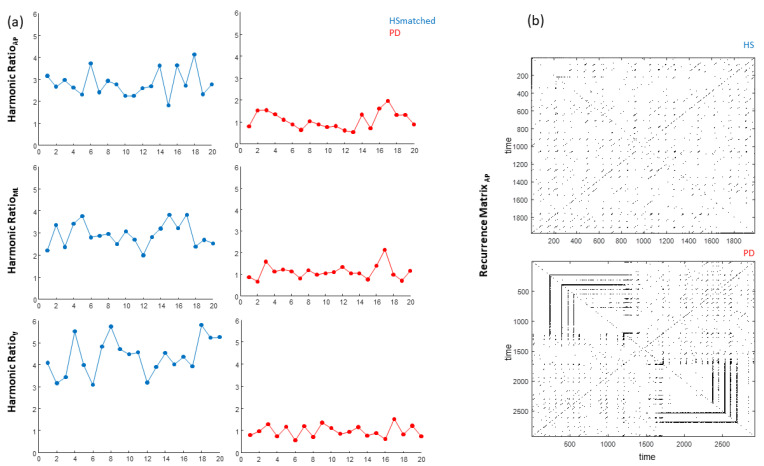
(**a**) Graphical representation of the Harmonic Ratios in the antero-posterior, medio-lateral, and vertical directions of a representative age-and-speed-matched healthy subject (blue) and a subject with PD at Hoehn and Yahr stage = 3 (red); (**b**) recurrence matrices in the antero-posterior direction of the same representative subjects.

**Figure 3 sensors-21-03449-f003:**
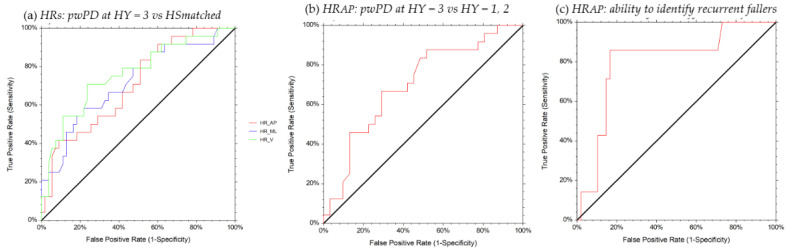
ROC curves for the HRs in identifying pwPD vs. HSmatched, pwPD at HY = 3 from milder HY and recurrent fallers. The red line represents the HR_AP_, the blue line the HR_ML_, and the green line the HR_V_.

**Figure 4 sensors-21-03449-f004:**
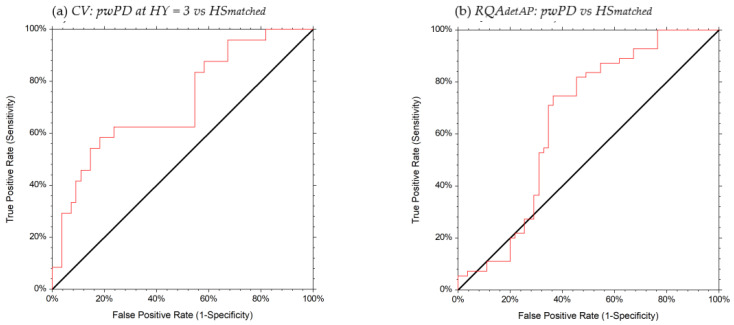
ROC curves of the CV (**a**) and RQAdetAP (**b**) in discriminating pwPD from HS_matched._

**Table 1 sensors-21-03449-t001:** Comparison of the gait stability indexes between subjects with PD and HS_matched_.

	HY 1	HY 2	HY 3	HSmatched
	Mean (SD)	Mean (SD)	Mean (SD)	Mean (SD)
HR_AP_	2.00 (0.55)	1.94 (0.51)	1.65 (0.34) * #	2.08 (0.67)
HR_ML_	1.82 (0.49)	1.84 (0.48)	1.63 (0.37) *	1.97 (0.51)
HR_V_	1.93 (0.53)	1.86 (0.41)	1.64 (0.37) *	2.05 (0.50)
RQA_recAP_	3.13 (2.04)	4.21 (2.91)	4.35 (4.10)	5.24 (3.54)
RQA_recML_	4.43 (4.83)	3.80 (4.08)	2.53 (3.77)	3.86 (3.31)
RQA_recV_	4.73 (5.50)	3.44 (4.65)	5.38 (7.62)	2.76 (2.25)
RQA_detAP_	23.20 (15.94) *	30.23 (17.89) *	28.37 (22.74) *	41.75 (27.07)
RQA_detML_	31.09 (26.35)	33.93 (26.27)	22.63 (25.10)	34.93 (25.38)
RQA_detV_	27.17 (28.83)	20.61 (20.05)	25.40 (23.67)	23.78 (16.04)
CV	39.36 (17.85)	35.56 (16.94)	40.92 (18.30) *	28.72 (14.08)
NJS_AP_	2261.67 (1387.79)	4157.58 (3465.20)	4061.89 (2839.03)	3512.49 (3047.39)
NJS_ML_	1169.38 (959.77)	1518.38 (1115.65)	1622.29 (1217.89)	1558.43 (1221.20)
NJS_V_	1127.09 (808.44)	1526.17 (1166.67)	1591.34 (1925.34)	1656.37 (1169.45)
LLE_AP_	0.53 (0.26)	0.49 (0.20)	0.60 (0.23)	0.53 (0.26)
LLE_ML_	0.64 (0.23)	0.58 (0.18)	0.63 (0.20)	0.54 (0.28)
LLE_V_	0.82 (0.28)	0.88 (0.21)	0.88 (0.21)	0.86 (0.29)

* significant differences between subjects with PD and HSmatched; #, significant differences between subjects with PD at HY 1,2 and HY 3; HY, Hoehn and Yahr disease staging classification system; HR, harmonic ratio; RQArec, recurrence quantification analysis, percent recurrence; RQAdet, recurrence quantification analysis, percent determinism; CV, coefficient of variation of the step length; NJS, normalized Jerk score; LLE, largest Lyapunov exponent.

**Table 2 sensors-21-03449-t002:** ROC curve and cutoff analysis results.

Gait Index	Subjects	AUC (95% CI)	OCP	Se (95% CI)	Sp (95% CI)	LR+	LR−	PTP+	PTP−
HR_AP_	HY = 3 vs. HS_matched_	0.71 (0.56–0.81)	≤1.50	0.42 (0.22–0.63)	0.91 (0.80–0.97)	4.58	0.64	67%	22%
	HY = 3 vs. HY = 1.2	0.70 (0.53–0.80)	≤1.50	0.46 (0.25–0.67)	0.87 (0.70–0.96)	3.55	0.62	73%	32%
	recurrent fallers (≥5)	0.80 (0.52–0.92)	≤1.50	0.85 (0.42–0.98)	0.83 (0.70–0.92)	5.14	0.17	77%	10%
HR_ML_	HY = 3 vs. HS_matched_	0.72 (0.56–0.82)	≤1.58	0.58 (0.36–0.78)	0.78 (0.65–0.88)	2.67	0.53	54%	19%
HR_V_	HY = 3 vs. HS_matched_	0.76 (0.61–0.86)	≤1.74	0.71 (0.49–0.87)	0.76 (0.63–0.87)	3.00	0.38	57%	14%
RQA_detAP_	All subjects with PD vs. HS_matched_	0.65 (0.53–0.75)	≤38.85	0.74 (0.61–0.85)	0.63 (0.49–0.76)	2.05	0.40	67%	29%
CV	HY = 3 vs. HS_matched_	0.72 (0.57–0.82)	≥38.06	0.58 (0.36–0.78)	0.81 (0.69–0.91)	3.21	0.51	58%	18%

AUC, area under the ROC curve; OCP, optimal cutoff point; Se, sensitivity; Sp, specificity; LR+, positive likelihood ratio; LR−, negative likelihood ratio; PTP+, positive post-test probability; PTP−, negative post-test probability; HR, harmonic ratio; RQA_detAP_. Recurrence quantification analysis, percent determinism in the antero-posterior direction; CV, coefficient of variation. of the step length.

## Data Availability

The data presented in this study are available on request from the corresponding author and stored in a password-protected PC located in the Department of Medico-Surgical Sciences and Biotechnologies, University of Rome Sapienza.

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
