# Peer review of "Ability of a Set of Trunk Inertial Indexes of Gait to Identify Gait Instability and Recurrent Fallers in Parkinson’s Disease"

_sensors, 2021, doi:10.3390/s21103449_

Round 1

Reviewer 1 Report

Castiglia et al. present a manuscript related to trunk inertial indexes of gait to identify gait instability and recurrent fallers in PD. The authors use wearable sensors to capture the data and then extract features from these data to identify gait instability and recurrent fallers in PD.

Falls are an important problem in PD, which can lead to significant injury and decreased quality of life. Thus, identifying highly-discriminant features related to postural and gait instability are of great benefit. However, there are several points that this reviewer feels the authors would need to address before this manuscript is suitable for publication.

1)  I think the hypothesis in the introduction needs to be focused. For example:

Does there really need to be 3 hypotheses?

Some gait stability indexes may have sufficient ability to discriminate between pwPD and HS, regardless of gait speed.” Why is this important? For example, something like the single-leg-stance-test can be used to assess postural instability; which is a simple and fast clinical test where gait speed is irrelevant.

Also, can the two hypotheses - “Some gait stability indexes may have sufficient ability to discriminate between pwPD and HS, regardless of gait speed.” and “The identified discriminant indexes may be correlated with the spatiotemporal and kinematic gait features of pwPD, regardless of gait speed” not be combined?

2) A range of m and τ should be reported, as well as the method used to estimate them. Also, I am not sure I agree with the statement higher Rec means more regular the dynamic structure. I can think of some counter examples where this may not be the case (e.g., during a freezing episode, or even the author’s Table 1). Also, given the authors were interested in the largest Lyapunov exponent, why didn’t they consider RQA(Lmax) as this variable as it inversely scales with the most positive Lyapunov exponent?

3) “To assess the ability of the identified indexes to discriminate between pwPD and HSmatched, classify pwPD into HY stages, and identify recurrent fallers,....”  This reviewer feels that the authors really need to justify the last point in particular if the goal is really to identify recurrent fallers. That is because a clinician can simply ask for fall history, and thus there would be no need for gait tests (which themselves poses some fall risk). If this data is from individuals without a fall history at assessment, who then became recurrent fallers, then this would be quite useful. However, I am not sure that is the case here. Also, for “classify pwPD into HY stages” it would need to be shown that this new approach offers some distinct advantage (e.g., faster, cheaper, …).

4) Additional details are needed throughout the manuscript. For example, “The mean gait speed of the included pwPD was 0.76 ± 0.23.” I assume m/s?

5) It appears the authors are strongly claiming that they are looking at matched groups. However, “gait speed (F (3, 106) = 2.49; p = 0.06)” is marginally significant. 

Author Response

Response to Reviewers.

We would thank the reviewers for their general judgments about our work and for their qualified, detailed, and constructive comments and suggestions that helped us in improving our work. Below, we have reported all our point-by-point responses to each reviewer. All changes and corrections within the text have been written in red color.

Reviewer 1

Castiglia et al. present a manuscript related to trunk inertial indexes of gait to identify gait instability and recurrent fallers in PD. The authors use wearable sensors to capture the data and then extract features from these data to identify gait instability and recurrent fallers in PD.

Falls are an important problem in PD, which can lead to significant injury and decreased quality of life. Thus, identifying highly-discriminant features related to postural and gait instability are of great benefit. However, there are several points that this reviewer feels the authors would need to address before this manuscript is suitable for publication.

We would like to thank the Reviewer for his valuable comments, which helped us to improve our manuscript.

1)  I think the hypothesis in the introduction needs to be focused. For example:

Does there really need to be 3 hypotheses? “Some gait stability indexes may have sufficient ability to discriminate between pwPD and HS, regardless of gait speed.” Why is this important? For example, something like the single-leg-stance-test can be used to assess postural instability; which is a simple and fast clinical test where gait speed is irrelevant.

We agree with the Reviewer that many clinical tests have been developed to assess unbalance in pwPD, most of them performing suboptimally or having been studied insufficiently in terms of clinimetric properties (Bloem et al., 2016). Some of them, such as the MiniBestTest (Franchignoni et al., 2010), which is a comprehensive tool that includes SLST, is able to identify unbalance and risk of falls in pwPD (Winser et al., 2019). However, these measures provide clinicians clinically useful tools to describe the status of the patients or their risk of falls, but they are not designed to provide quantitative measures as well as to describe the subtle alterations leading to falls in pwPD. Measuring the gait stability indexes through wearable gait analysis devices may allow to have more objective biomechanical markers directly reflecting the trunk behavior during gait, which has been described to be correlated with the history of falls (Cano-de-la-Cuerda et al., 2017,2020) and a predictor of gait quality improvement after rehabilitation in pwPD (Serrao et al., 2019). In this study, we aimed at observing if the subtle trunk behavior alterations during the gait of pwPD, as assessed through the trunk acceleration-derived gait stability indexes, could characterize the gait alterations of pwPD as respect to healthy subjects. For this purpose, it is necessary to exclude the effects of the gait speed on the other gait parameters, because most of them, including trunk acceleration, are speed-dependent and so gait speed may bias the differences between pwPD and healthy subjects (Fukuchi et al., 2019). In this study, we found that the alteration of the harmonic content of trunk acceleration during gait characterizes the gait of pwPD at worse disease status and characterizes the gait of pwPD who recurrently falls, excluding the gait speed as biasing factor. Moreover, the HR alteration in the AP direction is correlated with the trunk kinematic behavior as reflected by pelvic kinematics in this study. This result may provide clinicians a useful outcome measure and further suggests focusing on trunk and pelvic kinematics when designing rehabilitative interventions for improving the gait stability of pwPD. Notably, gait assessment using inertial sensor units is a fast and simple procedure, and the calculation of the significant indices may be implemented in the routinary gait assessment in rehabilitative contexts.

We better explained this issue in the revised “introduction” section and provided more supporting references, accordingly.

Specifically, we wrote:

(line 50-53)“Although many clinical tools have been developed to identify unbalance in pwPD, technology-based objective measures may improve the ability to capture motor behaviors to optimize treatment strategies.”

 And: (line 73-78) “Additionally, because the correlations between gait speed, age, and trunk accelerations have been well described and because walking stability is strongly influenced by gait speed in pwPD, which is a marker o gait alteration common to most neurological diseases, the value of the gait stability indexes as independent markers of gait alteration should be investigated, regardless of these confounding factors.”

Also, can the two hypotheses - “Some gait stability indexes may have sufficient ability to discriminate between pwPD and HS, regardless of gait speed.” and “The identified discriminant indexes may be correlated with the spatiotemporal and kinematic gait features of pwPD, regardless of gait speed” not be combined?

We combined the two hypotheses, as suggested. Specifically, we wrote:

(line 101-106) “Some gait stability indexes may have sufficient ability to discriminate between pwPD and HS and may be correlated with the spatio-temporal and kinematic gait features of pwPD, regardless of gait speed. ii) The identified discriminant indexes characterize the gait of recurrent fallers among pwPD as an expression of walking instability”

2) A range of m and τ should be reported, as well as the method used to estimate them. Also, I am not sure I agree with the statement higher Rec means more regular the dynamic structure. I can think of some counter examples where this may not be the case (e.g., during a freezing episode, or even the author’s Table 1). Also, given the authors were interested in the largest Lyapunov exponent, why didn’t they consider RQA(Lmax) as this variable as it inversely scales with the most positive Lyapunov exponent?

The ranges of m and τ and their estimation method with references have been added in the revised version of the manuscript, accordingly.

In our study, we excluded subjects with freezing of gait (FOG), as stated in the methods section, and we were interested in linear walking at steady state. FOG mainly appears during gait initiations, transitions, turning or dual task conditions, that were not the objects of our study. However, consider that we calculated the gait stability indexes from trunk acceleration signals. During a FOG episode, rapid oscillations of the trunk acceleration, corresponding to knee trembling, have been described, typically leading to an abrupt forward inclination of the trunk and falls (Okuma et al., 2014). Therefore, a less regular dynamic structure of trunk accelerations may be expected during FOG episodes too. Conversely, Afsar et al. (2018) observed higher percent determinism in more disabled subjects, but they calculated the RQA through a footswitch system, therefore the higher recurrency is attributable to the feet trembling during a FOG episode. As the Reviewer noticed, some of the RQArec indexes are higher in pwPD than HSmatched, but these differences are not significant, meaning that these variables were not characteristic of pwPD gait. Conversely, when the RQA values were lower than HS, this difference was significant, as for the RQAdetAP, meaning that the unpredictability of the dynamic system in the AP direction is a characteristic of the disease.

Regarding the calculation of the RQA (Lmax), we agree with the Reviewer that it would have been a very interesting parameter to study. However, in this study we were interested in observing the discriminative ability of the trunk-acceleration derived gait stability indexes mostly reported in the literature. To our knowledge, few studies reported a calculation of Lmax for gait data, reporting a very high number of strides to achieve reliability for this index (de Oliveira Assis et al., 2020; Riva et al., 2014). In this study, we did not reach the needed number of strides (>150) for a reliable assessment of Lmax, hence we believe that a specific study with longer walking tasks, such as during real life contexts, should be conducted to analyze its characteristics.

According to the Reviewer’s suggestions, we have now written in the Methods section: (line 202-210) “The embedding dimension “m” is the first recurrence parameter and is estimated using the nearest-neighbor method, which compares the distances between neighboring trajectories at successively higher dimensions. ‘‘False neighbors’’ occur when trajectories that overlap in dimension  are distinguished in dimension . As i increases, the total percentage of false neighbors decrease and m is chosen where this percentage approaches 0. False neighbor analysis was performed using values of Rtol = 17 and Atol =2. We considered a maximum embedding dimension of 10 (range: 2-10), and m = 5 was considered as the optimal embedding dimension”

And:

(line 216-222)“τ was calculated from the first minimum of the average mutual information (AMI) function, which evaluates the shared amount of information in bits between two data sets over a range of time delays. By choosing the first minimum of the AMI function, adjacent delay coordinates with a minimum of redundancy are provided. A range of τ between 7 and 18 was evaluated in this study. The time delay computed by the first minimum of the AMI considered as optimal was 10 samples.”

Moreover, we have now written, in the “Discussion” section:

(line 573-579)“Noteworthy, other acceleration-derived gait stability indexes, such as entropy measures or maximal longest diagonal in recurrence quantification analysis, that we did not consider in this study, can be calculated to assess human gait instability. However, being these indexes highly dependent on the number of recorded consecutive strides, other studies involving long-time gait series recordings are needed to analyze their ability to characterize the gait instability of pwPD.”    

3) “To assess the ability of the identified indexes to discriminate between pwPD and HSmatched, classify pwPD into HY stages, and identify recurrent fallers,....”  This reviewer feels that the authors really need to justify the last point in particular if the goal is really to identify recurrent fallers. That is because a clinician can simply ask for fall history, and thus there would be no need for gait tests (which themselves poses some fall risk). If this data is from individuals without a fall history at assessment, who then became recurrent fallers, then this would be quite useful. However, I am not sure that is the case here. Also, for “classify pwPD into HY stages” it would need to be shown that this new approach offers some distinct advantage (e.g., faster, cheaper, …).

We agree with the Reviewer that a longitudinal study should have been conducted to evaluate the ability of the gait stability indexes to predict future recurrent falls. In this study, we aimed at detecting the ability of the gait stability indexes to characterize the gait of pwPD who fall recurrently. To optimize clinical and rehabilitative interventions, clinicians could be interested in understanding the gait alteration characterizing recurrent fallers, rather than the mere rate of falls. Regarding the classification into HY stages, our study did not aim at introducing a new disease classification system, rather providing a description of the gait-acceleration behavior according to the progression of the disease. We modified the introduction and methods section, accordingly.

Specifically, we wrote in the “Introduction” section:

(line 97-98) “assessing the ability of each identified index to characterize the gait of pwPD who fall recurrently, compared with pwPD who do not fall recurrently”;

And, in the “statistical analysis” paragraph:

(line 324-328) “To assess the ability of the identified indexes to discriminate between pwPD and HSmatched, characterize the gait of pwPD according to the HY stages, and characterize the gait of recurrent fallers, ROC curves were plotted and AUCs were calculated with the presence of PD, HY stages, and a reported number of falls ≥ 5 as anchors, respectively.”

We also modified the terms “identify fallers” into “characterize the gait of recurrent fallers” throughout the manuscript.

4) Additional details are needed throughout the manuscript. For example, “The mean gait speed of the included pwPD was 0.76 ± 0.23.” I assume m/s?

The unit of measurement m/s for gait speed has been added, accordingly (line 356).

5) It appears the authors are strongly claiming that they are looking at matched groups. However, “gait speed (F (3, 106) = 2.49; p = 0.06)” is marginally significant.

Please, consider that this is the overall result of an ANOVA procedure, considering not only the differences between the pwPD and HS, but also the differences between the HY subgroup in pwPD, which could have lowered the significance level. Considering the Reviewer’s observation, we checked for the post-hoc comparison in the ANOVA procedure, and we did not find any significant difference between the HY subgroups of pwPD and HS, neither between the HY subgroups themselves. We reported the post-hoc results in the revised results section.

Specifically, we wrote:

(line 363-369) “No significant overall differences between group means were identified for age (F (3, 105) = 2.81; p = 0.29) or gait speed (F (3, 106) = 2.49; p = 0.06) after the matching procedure. Post-hoc analysis revealed no significant differences for age (HY=1 vs HSmatched: p = 1.00; HY = 2 vs HSmatched: p = 1.00; HY =3 vs HSmatched: p = 0.09; HY = 1 vs HY = 2: p = 1.00; HY = 1 vs HY = 3: p = 0.11; HY = 2 vs HY = 3: p = 0.25) and gait speed (HY=1 vs HSmatched: p = 1.00; HY = 2 vs HSmatched: p = 1.00; HY =3 vs HSmatched: p = 0.29; HY = 1 vs HY = 2: p = 1.00; HY = 1 vs HY = 3: p = 0.12; HY = 2 vs HY = 3: p = 0.21) between the HY stage subgroups and HSmatched, neither between the HY subgroups.”

Reviewer 2 Report

Paper on an interesting and little-researched topic.

However, it requires a few corrections:

ABSTRACT:

There are too many abbreviations and too much data in the abstract. As a result, it is not readable. It needs to be corrected.

Introduction :

Please write more about wearable sensors and add more references about them.

Materials and Methods:

Please give information on what basis the degenerative changes in the joints (hip, knee, ankle joint) were excluded in all patients – did You made all Patients  X-rays ? These changes may affect gait.

Please add a reference to: Mini-Mental State Examination, Beck Depression Inventor.

What is UPDRS-IIIstandard ? Description please.

Discussion:

Please correct the reference format in line 487-490.

I believe that it is easier to use a pedobarographic platform with a balance assessment to assess the balance and detect people at risk of falls accidents.Please discuss it.

Line 504-505, please add references.

Author Response

Response to Reviewers.

We would thank the reviewers for their general judgments about our work and for their qualified, detailed, and constructive comments and suggestions that helped us in improving our work. Below, we have reported all our point-by-point responses to each reviewer. All changes and corrections within the text have been written in red color.

Reviewer 2

Paper on an interesting and little-researched topic.

We would like to thank the Reviewer for his valuable comments.

However, it requires a few corrections:

ABSTRACT:

There are too many abbreviations and too much data in the abstract. As a result, it is not readable. It needs to be corrected.

Complying with the words number restriction, we modified the Abstract section and spelled-out many abbreviations, accordingly.

Introduction:

Please write more about wearable sensors and add more references about them.

We added further information about wearable sensors and added more references, accordingly.

Specifically, we wrote:

(line 81-89) “Based on the inverted pendulum model of gait, mobile inertial measurement units (IMU), which embed accelerometers and gyroscopes, are able to objectively capture the ability to control the body center of mass while moving the base of support, resulting in an effective tool to monitor dynamic balance during gait. Ideally, by using several combined IMUs to analyze a subject’s gait, the overall accuracy of the analysis would improve, but this gain would be offset by the wearability burden. Conversely, a single lumbar-mounted IMU provides sensitive information on the gait of pwPD and allows clinicians to monitor the gait of pwPD also in free-living conditions.”

Materials and Methods:

Please give information on what basis the degenerative changes in the joints (hip, knee, ankle joint) were excluded in all patients – did You made all Patients  X-rays ? These changes may affect gait.

We would like to thank the Reviewer for having pointed out the lack of information on exclusion criteria. In this study, a clinically based assessment of osteoarthritis according to OARSI criteria was performed before the enrollment in the study. We did not submit all patients to X-rays, but we excluded subjects complaining with hip, knee or ankle joints and reporting limitations on range of motions or visible anatomical alterations (such as Heberden’s nodules, palpable warmth of the knee, patellofemoral grinding, joint line tenderness, bony enlargement). We specified it in the revised version of the manuscript.

Specifically, we wrote:

(line 122-126) “(iii) orthopedic and/or other gait-influencing diseases, such as other neurological diseases, clinically defined osteoarthritis, or total hip joint replacement. Particularly, subjects referring pain in hip or knee joints and reduced range of motion on internal hip rotation, hip flexion and visible anatomic alterations of the joints were excluded.”

Please add a reference to: Mini-Mental State Examination, Beck Depression Inventor.

References for MMSE and BDI have been added, accordingly (line 121-122).

What is UPDRS-IIIstandard ? Description please.

We modified the sentence and spelled out the UPDRS-III abbreviation, accordingly. The term “standard” was a typo. We thank the Reviewer for having pointed it out.

We have now written:

(line 295-297) “The severity of PD was assessed using the HY disease staging system and the motor examination section of the Unified Parkinson’s Disease Rating Scale (UPDRS-III)”

Discussion:

Please correct the reference format in line 487-490.

The reference format has been corrected, accordingly (line 531-534).

I believe that it is easier to use a pedobarographic platform with a balance assessment to assess the balance and detect people at risk of falls accidents. Please discuss it.

The aim of this study was to assess the ability of the analyzed gait stability indexes to characterize the gait of pwPD and the gait of recurrent fallers among such a population. However static postural sway as assessed through a pedobarographic platform has proven to be able to parallel the disease progression in pwPD and to highlight the reactive postural control deficits (Frenklach et al., 2009), pwPD mainly fall during dynamic tasks and exhibit restrictive trunk control during walking (Jehu et al., 2018) with impaired head and trunk stability during gait (Cole et al. 2017).  Because pedobarographic platforms cannot provide information on gait quality and cannot provide deep information about trunk acceleration patterns during walking, the use of the widespread and easy-to-use IMUs for gait analysis appears better suitable to our aims than pedobarographic platforms.

 Line 504-505, please add references.

The needed references have been added, accordingly.

Round 2

Reviewer 1 Report

The authors have addressed the majority of my comments. I still not sure I agree that higher Rec always means more regular the dynamic structure. In fact, the reference to the Afsar paper and DET seems to be confusing since REC was fixed and DET decreased. 

Please proof read for typos: e.g., "which is a marker o gait alteration common to most neurological diseases"

Author Response

Reviewer 1

The authors have addressed the majority of my comments. I still not sure I agree that higher Rec always means more regular the dynamic structure. In fact, the reference to the Afsar paper and DET seems to be confusing since REC was fixed and DET decreased. 

We would like to thank the Reviewer for his attention in remarking on this point. We realized that in the previous reply we misinterpreted the data from Afsar et al. (2018).  Although they did not report data on percent recurrence, Afsar et al. reported a decrease of percent determinism in more disabled pwPD, which is consistent with the results of our study. However other Authors (Labini et al., 2012, Tamburini et al., 2018; Weber et al., 1994) reported that subjects with better walking ability exhibit high regularity (i.e. high RR and DET), we agree with the Reviewer that this does not always mean better gait behavior. In particular conditions, such as in subjects who experienced a concussion (Howell et al., 2020), the higher predictability may represent an expression of the alteration in the adaptability to gait perturbations. According to the Reviewer’s observation, to avoid equivocal interpretation, we modified the sentence in the Methods section, line 236-237.

Specifically, we wrote: “The higher the RQArec and RQAdet values, the higher the predictability of the system”.

Please proof read for typos: e.g., "which is a marker o gait alteration common to most neurological diseases"

We proofread the manuscript for typos, accordingly.